# Invitation appeals and STEM academic scientists research participation: Findings from six survey experiments

Tipeng Chen[iD][1]*, Timothy P. Johnson[2], Jinghuan Ma[1], Ashlee Frandell[3], Lesley Michalegko[1], Eric W. Welch[1]*

**1** Center for Science, Technology and Environmental Policy Studies, School of Public Affairs, Arizona State University, Phoenix, Arizona, United States of America, **2** Department of Public Policy, Management, and Analytics, University of Illinois at Chicago, Chicago, Illinois, United States of America, **3** School of Public Policy and Leadership, University of Nevada Las Vegas, Las Vegas, Nevada, United States of America

* tpchen@asu.edu (TC); ericwelch@asu.edu (EW)

## Abstract

Survey research is a primary method used to investigate the opinions, perceptions and behaviors of academic scientists. However, little is known about the most successful appeal strategies for eliciting survey participation from these busy, highly educated professionals. Drawing on leverage-salience theory, this study examines the impacts of two sets of invitation appeals—information and representation appeals—on survey response rates among academic scientists in four STEM fields employed at U.S. R1 universities. Findings from six randomized experiments show that the effectiveness of both sets of invitation appeals is mixed and context-dependent, varying based on the polarization and relevance of survey topics, STEM academic scientists' career stage, and their prior interactions with survey administrators. Specifically, self-representation appeals are most effective for polarized topics when recipients have low community affiliation. Less detailed information appeals are more successful when asking about low relevance topics, particularly for recipients with greater demands on their time, while more detailed information is effective for highly relevant and polarized topics. Additionally, invitations containing more detailed information are effective for first-time recipients in survey panels. This complexity reinforces the importance of designing effective outreach strategies to account for survey topics and recipient characteristics.

## Introduction

Surveys have long been utilized to study the community of academic scientists in science, technology, engineering, and mathematics (STEM) fields, a practice that dates back more than half a century [1–3]. Research in this area has explored a wide

**Data availability statement:** The de-identified data files are available from the Harvard Dataverse: https://doi.org/doi:10.7910/DVN/7YHUBI

**Funding:** The authors received no specific funding for this work.

**Competing interests:** The authors have declared that no competing interests exist.

range of topics relevant to STEM academic scientists, ranging from their personal life to professional activities, to expert opinions. Scholars have examined the dynamic motivations and beliefs driving STEM academic scientists' behaviors and opinions [4–6], the socioeconomic determinants for STEM academic scientists' career choices, discovery and innovation activities [7–10], their various professional networks (e.g., collaboration, competition, and mentorship) [11,12], and their university/work environments (e.g., representation, inclusion, and equity) [13–15].

Surveying STEM academic scientists also holds considerable promise for enhancing science communication. The traditional deficit model of science communication assumes that scientific knowledge flows one-way from experts to the public. Under this model, communication failures—especially when the public distrusts science or holds "anti-science" views—are often attributed to media misinterpretation and public ignorance [16,17]. However, by aggregating the opinions of STEM academic scientists—those with the most widely recognized expertise in their fields—surveys provide a revised approach for science communication [18]. Survey administrators publish scientific opinions in a more accessible and understandable format for public authorities and the public through digital platforms [19], bypassing the social media's potential role in misinterpreting science and polarizing audience views on science [20,21]. Institutions such as the Pew Research Center and the National Center for Science and Engineering Statistics (NCSES) routinely conduct surveys to synthesize and publish the views of leading STEM academic scientists [22–25]. These efforts enhance science communication by providing valuable assessments, insights, and consultations that inform policy decisions and influence individual behavioral changes [18,26–28].

Surveys further foster a two-way dialogue by recognizing STEM academic scientists—one of the most educated subgroups of the public—as both contributors to and recipients of scientific knowledge. By capturing diverse and, at times, divided perspectives within the STEM academic community, surveys contribute to constructive public debate—especially when knowledge about reality is uncertain, underdeveloped, and/or the issues are politicized [2,29,30]. Examples include contentious topics such as climate change [31,32], emerging biotechnology [28,33], women's abortion rights [34], and vaccine safety [35]. By presenting diverse scientific opinions, surveys help structure information environments where the public, policymakers, and the private sector can carefully consider multiple sides of a controversial issue [36,37] and researchers have opportunities to refine their work for a broader social consensus [38,39].

In the face of declining survey response rates and politicization of science communication [4,30,40,41], understanding how to effectively engage STEM academic scientists in survey research and tailor surveys specifically to them becomes important. Through a series of online survey experiments, this study examines the effectiveness of two sets of appeal strategies designed to increase response rates among STEM academic scientists. The study focuses on those employed by Carnegie-designated Research Extensive and Intensive (R1) universities in the United States (U.S.) working in four STEM fields – biology, civil and environmental engineering, geography,

public health. One set of appeals focuses on levels of information provided to STEM academic scientists at the time they are invited to participate. Another set of appeals varies representation, with one appeal condition emphasizing participation as representative of the scientific community and another emphasizing the importance of expressing personal opinions. This study contributes to our understanding of survey appeal strategies that may influence STEM academic scientist's willingness to participate in online survey research.

## Literature, theory, and hypothesis

Initial contact and effective communication are important to elicit cooperation in web surveys [42–44]. Low response rates can reduce effective sample sizes, which limits statistical power and increases the risk of nonresponse bias [45]. For a number of years, researchers have been examining how to better elicit respondent cooperation with online surveys by testing various strategies for developing effective communications [46], such as offering monetary incentives [47], designing persuasive invitation messages [48], and sending optimal notification and follow-up reminders [49]. However, these design strategies may not all be applicable to STEM academic scientists. For example, monetary incentives are known to be less effective for highly educated populations [50]. Due to their unique characteristics, STEM academic scientists may exhibit survey participation behaviors that differ considerably from those of the general population, presenting both opportunities and challenges in survey design and implementation.

In terms of opportunities, response rates of STEM academic scientists tend to be higher than those of the general public due to several factors. First, STEM academic scientists are generally prosocial and sympathetic to improving public knowledge about science and scientific activities [38,51]. There is growing encouragement for STEM academic scientists to share their expertise and opinions beyond their institutions by active engagement in science communication within the public domain [17,52–54]. Second, STEM academic scientists may be more familiar with survey methodologies and their rights as human research subjects and have a clearer understanding of the risks and benefits associated with participating in surveys. Their digital literacy enables them easier access to online surveys, and they may have empathy with fellow academic scientists and believe in reciprocity to help other researchers [42]. Third, data from STEM academic scientist surveys can be linked with Big Data relevant to science behaviors (e.g., bibliometric data) to produce research outputs that would be impossible with either data source individually [55,56].

Meanwhile, some work-related characteristics of STEM academic scientists raise challenges for researchers when recruiting them for surveys. For one, the growing demand for STEM academic scientists to share their opinions has potentially led to an increasing number of survey invitations directed toward them [23–25]. This surge in survey requests contributes to increased survey fatigue among STEM academic scientists [57]. Moreover, given their various roles and obligations (e.g., research, teaching, mentoring, and service), STEM academic scientists face email fatigue due to the constant influx of emails from both internal institutions and various external sources [58,59], making them less likely to notice or be interested in online survey invitations. Also, compared with the general public, STEM academic scientists are highly educated professionals with domain-specific expertise and experience, making them relatively small in number, exceptionally busy, and less approachable [44,46]. Moreover, STEM academic scientists are a heterogeneous population, differing in academic positions (e.g., tenured, tenured-tracked, clinical, and teaching), rank (e.g., assistant, associated, and full), professional experience, and disciplinary backgrounds (e.g., biology, chemistry, etc.). Although biographical and demographic data on STEM academic scientists are publicly available, allowing researchers to compare and weigh their observable respondents with a reliable sample frame, such a heterogeneous group makes it difficult for researchers to obtain a representative sample of this target population.

Dillman and colleagues emphasize the importance of Tailored Design methods to customize survey procedures and instruments to minimize error and boost participation according to the characteristics of the target population [60]. This approach considers factors such as survey topic, sponsor, available resources, and timelines. Following the ideas of Tailored Design, some research investigates strategies to improve response rates specifically among educated professional

groups such as STEM academic scientists, physicians, and politicians, particularly in online surveys [42,44,61,62]. These strategies typically fall into three categories: incentives, notifications, and appeals.

Incentives for survey participation can be monetary or nonmonetary, though their effectiveness varies. Monetary incentives differ in amount (ranging from under $2 to more than $50), format (cash or lottery), and timing (prepaid or post-survey completion) [62,63]. Evidence suggests that prepaid cash incentives are more effective at improving response rates than other formats, and higher amounts generally yield better participation [61–63]. Nonmonetary incentives, including small tokens such as pens, have shown limited effectiveness for increasing response rates among professionals [61,62]. While monetary incentives sometimes enhance response rates for these educated professionals, they also pose challenges [64]. Some public sector institutions prohibit their employees from accepting compensation, and small monetary amounts may be perceived as insulting [44,65]. Use of incentives also obviously increases the cost of collecting survey data. These challenges highlight the tradeoff of using incentives to boost response rates while considering ethical and cultural sensitivities for educated professionals such as STEM academic scientists.

A second common strategy to boost survey participation of educated professionals is sending notifications to sampled individuals, including pre-survey alert letters and follow-up reminders. Alert letters inform participants about the survey's purpose, rationale, sponsoring institution, and expected delivery date [66]. For example, Frandell and colleagues found that pre-notifications increased response rates by about 3% in experiments conducted across three surveys targeting STEM academic scientists at U.S. R1 universities [42]. Similarly, Hart and colleagues reported a 4% increase in survey participation among clinical professionals following pre-notifications [67]. Follow-up reminders are another effective notification method, involving repeated contacts with non-respondents to encourage their participation. Reminders often include details about the survey's closing date and solutions for technical issues with electronic surveys. Guise and colleagues, in a non-experimental study among clinical professionals, found that repeated follow-ups could improve response rates [68]. Follow-up reminders are particularly helpful for highly educated professionals who may overlook initial invitations due to high email volumes or who become more available after the initial invitation period. While notifications are effective, they also present challenges [69]. Nonresponse may reflect implicit refusals, and excessive reminders can lead to survey fatigue, annoyance and/or a hostile survey climate, potentially reducing response quality for current and future surveys. Additionally, repeated follow-ups increase survey administration costs, and the marginal gains in response rates diminish with each subsequent reminder.

Compared to incentives and notifications, appeals offer a more cost-effective and labor-efficient strategy. Appeals act as customizable nudges that can be tailored to respondents' characteristics, the survey's purpose, and its topic. Research has examined the effects of multiple types of appeals that have been made as part of survey invitations. Appeals examined emphasize the interests of multiple parties, including the respondent, the greater public, and in a few cases, the investigator [70–73]. Appeals highlighting the interests of individual respondents are labeled authority [74,75], egoistic [72], and importance of respondent [71] appeals. In contrast, those appeals representing the interests of the broader public are labeled affiliation [75], altruistic [72], science [76], and social utility [71,77] appeals. The evidence available is mixed, with some experiments suggesting egoistic appeals are most effective [70,71,78], and others finding altruistic [76,79] or help-the-researcher appeals [80] to be more effective, and still others document no differences across appeal conditions [46,71,72,74,75,77]. Finally, some find no main effects of appeal type but report significant interactions between appeal and other variables, including the type of organization making the appeal [81], and the cultural background of respondents [75].

Although prior studies have explored the use of appeals to increase survey participation among educated professionals, such as teachers [75], physicians [74], and nurses [71], very little research exists that evaluates the relative effectiveness of various appeal types as part of surveys focusing on STEM academic scientists. These scientists possess unique work characteristics, such as time flexibility, familiarity with research ethics, appreciation of research activities, and understanding of survey methodology, which may shape their responses to survey invitations differently. Thus, it is

important to understand how different participation appeals influence STEM academic scientists' cooperation. Recognizing that traditional appeals—egoistic, altruistic, and help-the-sponsor—often fail to yield high response rates, this study investigates two sets of alternative appeals: (1) appeals emphasizing information detail and (2) appeals emphasizing type of representation.

Grounded in leverage-salience theory [43,82], we develop hypotheses about how these two sets of appeals may influence STEM academic scientists' propensity to participate in surveys. Salience is a construct referring to the extent to which specific attributes of a survey are perceived as motivating by potential survey respondents. Leverage-salience theory posits that individuals are more inclined to participate in surveys when they perceive certain attributes of the survey invitation as important (salient). Salience varies based on the targeted population's characteristics; for example, lottery incentives may appeal to low-income individuals [83], while altruistic appeals resonate more with prosocial individuals [48]. By emphasizing survey attributes that align with recipients' specific concerns and preferences, their likelihood of participation in surveys increases.

The first set of appeals operationalize the salience of information by varying the amount of detail provided about the survey topic as part of the invitation. We hypothesize that invitations with less information will be more salient with STEM academic scientists and effective for increasing response rates for several reasons. First, STEM academic scientists are accustomed to concise, precise communication in academic settings. Even though STEM academic scientists can process detailed information efficiently, excessive details perceived as providing marginal benefit may annoy scientists, decreasing their propensity to respond to the survey. Lengthy invitations with unnecessary details further increase survey complexity and ambiguity, reducing readability and potentially leading to concern about the quality of the survey and the qualifications of the research team. Second, while additional details may elaborate on the survey topic, they do not necessarily enhance its salience. Scientists can understand the purpose and significance of a survey from brief descriptions. Third, scientists are time-sensitive and thus value their time highly. Similarly, detailed invitations require more time to read and process, which may deter participation, especially among STEM academic scientists who often experience email fatigue [58–59]. In summary, the salience for STEM academic scientists lies in the clarity and brevity of the invitation, not the volume of information.

*Hypothesis 1: STEM academic scientists receiving less detailed survey invitations are more likely to respond than those receiving more detailed invitations.*

The second set of experiments use an integrative conceptualization of appeal design, operationalizing representation salience through two types of representation appeals that are most often examined in the literature. STEM academic scientists, on the one hand, might be invited to participate via appeals to self-representation that emphasize the importance of their professional expertise and individual voice. Such appeals align with egoistic appeals that emphasize individual respondent authority, knowledge and expertise [71,74,75,79–81]. Alternatively, appeals might be made to respondents as being representatives of the greater scientific community–a community-representation appeal. These types of appeals build on social identity theory that prioritizes community identity as a key motivator of prosocial behaviors, such as survey participation, that benefit the community [84]. These types of messages are similar to the emphasis placed on social and community benefits of participation typically found in altruistic appeals [70,77,79].

As noted earlier, findings from the literature on invitation appeals have been mixed, with some results suggesting that the salience of representation appeals is context-based [75,81], depending on other characteristics of survey design and/or the targeted population. Thus, the characteristics of the STEM academic scientific community also matter for any observed response differences between self-representation and community representation appeals. We hypothesize that self-representation appeals are more salient with STEM academic scientists than community-representation appeals. STEM academic scientists are more responsive to self-representation appeals, as they are generally more comfortable expressing their personal opinions than representing or speaking for the greater scientific community where disagreements are common. The scientific community is inherently heterogeneous, varying by disciplines, career stages, research

activities, capacities, and research endowments [85]. This heterogeneity means that the behaviors, research processes, and personal opinions of individual scientists often fail to represent the broader scientific community. Furthermore, scientific debates on contentious issues are both common and openly conducted. For instance, epidemiologists have engaged in transparent and vigorous discussions about COVID-19 social distancing policies and vaccine distribution on social media [86]. Most scientific arguments require further scrutiny, and issues with broad consensus among experts remain rare [36]. Science communication, grounded in critical objectivity, prioritizes evidence derived from rigorous scientific methods—such as replicable experiments and falsifiable hypotheses—over authoritative or popular opinions [87,88]. Because academic training emphasizes critical objectivity, STEM academic scientists tend to approach generalizations of their opinions with caution. They may perceive community-representation appeals as potentially introducing bias into survey results, leading to hesitation in responding to such appeals. Thus, STEM academic scientists' preference for individual expression over representing others likely makes them more inclined to respond to invitations framed with appeals that emphasize their personal perspective.

*Hypothesis 2: STEM academic Scientists receiving self-representation appeals are more likely to respond than those receiving community-representation appeals.*

## Method

### SciOPS and sampling strategy

This study draws on six survey experiments conducted by the SciOPS (Scientist Opinion Panel Survey) survey panel, a science communication platform developed by the Center for Science, Technology and Environmental Policy Studies at Arizona State University. SciOPS, consisting of a survey panel of randomly selected academic scientists in U.S. R1 universities, connects society with the scientific community by collecting and sharing the broadly representative opinions of U.S.-based STEM academic scientists on timely, critical science and technology issues. Further information on SciOPS surveys is available at: https://www.sci-ops.org/.

The survey experiments were embedded within six SciOPS surveys covering various topics: (1) COVID-19 Survey Wave 2; (2) COVID-19 Survey Wave 4; (3) Public Trust in Science Survey; (4) Survey of Scientists' Perceptions of Surveys; (5) Women's Health Survey; and (6) Vaccine Survey. All the above surveys followed a two-stage sampling and administration process. Table 1 summarizes the key features of these surveys.

**First stage.** Since 2020, SciOPS has built a pilot sample frame of approximately 12,000 academic scientists in four STEM fields: biology, geography, civil and environmental engineering, and public health. SciOPS used probability sampling to randomly select R1 universities across the U.S. (see S1 Table in Supporting Information). The research team then collected the names and contact information of tenured and tenure-track faculty (assistant, associate, and full professors) and PhD-holding non-tenure track researchers from publicly available websites of sample departments. In 2021, two groups of random samples from this pilot sample frame were drawn separately for three appeal experiments embedded in two COVID-19-related surveys (Waves 2 and Wave 4) and a Public Trust in Science Survey.

**Second stage.** In 2022, SciOPS launched its first panel member recruitment campaign by sending four rounds of invitations to STEM academic scientists in the pilot sample frame. Panel members were informed they would be invited to complete two surveys annually and would receive survey results and an annual certification of recognition. The campaign resulted in 986 eligible academic scientists joining the panel, yielding a recruitment rate (RECR) of 7.7% [89]. Following recruitment, we conducted three surveys with SciOPS panel members. The Survey of Scientists' Perceptions of Surveys and Women's Health Survey each used a random sample of all panel members. The Vaccine Survey targeted all biologists and public health faculty within the SciOPS panel. Two information appeal experiments and one representation appeal experiment were conducted as part of these surveys.

**Table 1. Description of survey experiments.**

| Survey | Experimental design | Dates | Topic | No. of essential survey questions[a] | Sampling strategy | Response rate |
|---|---|---|---|---|---|---|
| COVID-19 Survey Wave 2 | Representation Appeal, two conditions | May – June, 2021 | The impact of COVID-19 policies on research and personal life | 23 | A random sample from SciOPS initial sample frame including both non-SciOPS panel members and panel members | 16% |
| COVID-19 Survey Wave 4 | Information Appeal & Representation Appeal, a two-by-two experiment | May – June 2024 | The longer-term impact of COVID-19 policies on research and personal life | 21 | The same sample frame as COVID-19 Survey Wave 2 | 8.8% |
| Public Trust on Science Survey | Representation Appeal, two conditions | September – October, 2021 | Contributors to public's trust in science | 13 | A random sample from SciOPS initial sample frame including both non-SciOPS panel members and panel members | 20% |
| Survey of Scientists' Perceptions of Surveys | Information Appeal, three conditions | October – December, 2022 | Attitudes towards and experience with participating surveys | 7 | A random sample from SciOPS panel members | 34% |
| Women's Health Survey | Representation Appeal, two conditions | December 2022 – January, 2023 | Women's reproductive health | 11 | A random sample from SciOPS panel members | 37% |
| Vaccine Survey | Information Appeal, three condition | March – May, 2023 | Vaccine hesitancy and risk communication | 14 | Biologists and public health faculty in SciOPS panel members | 38% |

Note.

[a]Essential survey questions are the questions every respondent can see as they proceed through the survey, regardless of their answers to previous questions. This excludes items displayed conditionally through skip logic.

All surveys prior to 2022 were conducted using Sawtooth Software®, while those from 2022 onward used the Nubis® software system. Surveys were administered online in English. Table 1 shows that surveys limited to SciOPS panel members achieved higher rates (around 35%) than those including non-panel members (lower than 20%), and survey fatigue was evident in COVID-19 Survey Wave 4, which had a lower response rate than Wave 2 despite using the same sample frame. Fig 1 shows the timeline of SciOPS development and its six embedded survey experiments.

### Randomized controlled trials design

**Treatment condition.** The information appeal experiment tested the effect of varying the volume of information about survey topics in the invitation to potential respondents. This experiment was included in the Survey of Scientists' Perceptions of Surveys and the Vaccine Survey. Invitations included one of three conditions: no information about the survey topic, some information (approximately 45 words), or detailed information (approximately 90 words). Fig 2 illustrates the operationalization of these conditions in the Survey of Scientists' Perceptions of Surveys while Fig 3 presents the corresponding operationalization in the Vaccine Survey. The complete set of the email invitations used in these experiments is provided in Supporting Information as S1 and S2 Appendices.

The representation appeal experiment tested the effectiveness of self-representation vs. community-representation appeals. This experiment was conducted in the COVID-19 Survey Wave 2, the Public Trust in Science Survey, and the Women's Health Survey. In the self-representation condition, invitations encouraged STEM academic scientists to participate

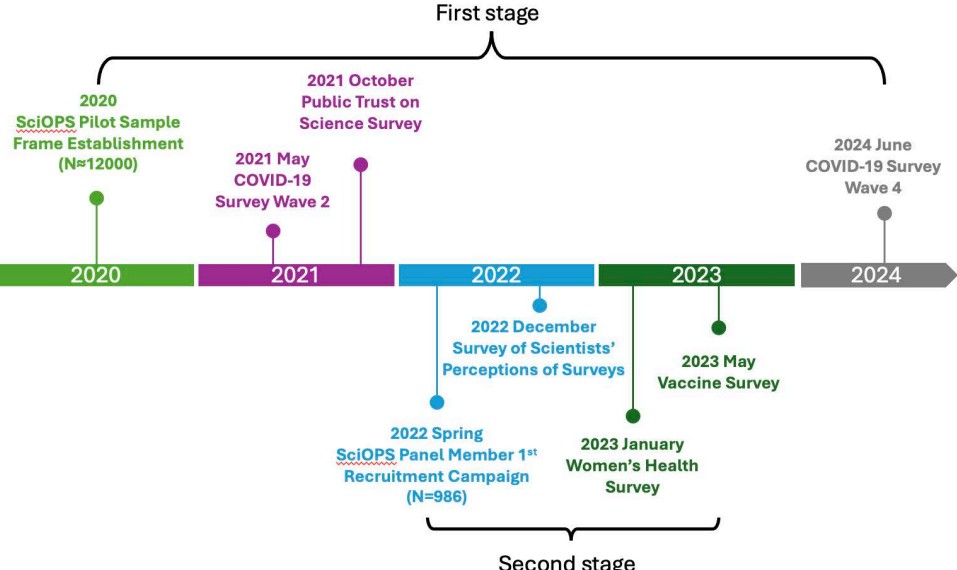

**Fig 1. Timeline of SciOPS survey experiment.**

---

Subject Line: SciOPS panelists survey practices

Dear Dr. [Surname only]:

Thank you for joining SciOPS (Scientist Opinion Panel Survey), a nationally representative panel of the Science, Technology and Innovation (STI) community.

I am writing to ask for your help with a national survey of the STI community concerning **researchers' survey practices**. We are interested in your opinions about our survey methods!

[Randomized (1) no information appeal: " "/

(2) some information appeal: "This particular survey will ask about your personal survey experiences, your opinion regarding important topics for future study and related issues. It will help us in the future to better design and focus SciOPS for communicating your opinions and experiences to the scientific community, policymakers, journalists and other stakeholders."

(3) much information appeal: "SciOPS is an important tool for engaging the scientific community and ensuring their voices are represented in the policy-making process. It also seeks to monitor trends over time and communicate to the public the importance and value to society of scientific research. It does so using a rigorous, replicable and transparent survey process. This particular survey will ask about your personal survey experiences and opinions regarding them. It will help us in the future to better design and focus SciOPS for communicating your opinions and experiences to the public, policymakers, journalists and other stakeholders."]

The questionnaire is short and should only take about ten minutes of your time.

**Fig 2. Information appeal experiment in survey of scientists' perceptions of surveys.**

Subject Line: SciOPS Survey Invitation – Vaccines and Public Policy

Dear Dr. [Surname only]:

I am writing to ask for your help with a national survey of the STI community concerning **the role of scientists in communicating and developing public policy about vaccines** that is being conducted by the SciOPS (Scientist Opinion Panel Survey) team.

[Randomized (1) no information appeal: " " /

(2) some information appeal: "The survey will ask you about your personal opinions regarding vaccines, vaccine hesitancy, and relevant public policy. Your responses will help us better understand how scientists and the scientific community view these important questions and help inform public debate on these topics." /

(3) much information appeal: "There is currently considerable public interest and debate about this topic! The survey will ask you about your personal opinions and beliefs regarding vaccines, vaccine hesitancy and vaccine messaging, the effectiveness of relevant public policy tools, and your institution's response to recent vaccine controversies. Your responses will help us better understand how scientists and the scientific community view these important questions and how the opinions of the scientific community compare to those of the general population. Findings will also help inform public debate on these topics."]

The questionnaire is short and should only take about ten minutes of your time.

**Fig 3. Information appeal experiment in vaccine survey.**

Subject Line: How has COVID-19 affected [Randomized (1) self-representation appeal: "your" / (2): community-representation appeal: "academic "] research?

Dear Dr. [Surname only]:

I am writing to ask for your help with a national survey of scientists and engineers concerning the effects of COVID-19 on [Randomized (1) self-representation appeal: "your personal research. You are part of a random sample that has been chosen to complete a brief questionnaire. Your opinion and experiences are needed to inform further understanding about this important topic." / (2) community-representation appeal: "academic research. You are part of a random sample that has been chosen to represent the scientific community by completing a brief questionnaire concerning this important topic."]

As you know, the COVID-19 has affected most areas of life in the US. Your response is vitally important for learning about the experience of scientists such as yourself. The survey will ask you about COVID-19 impacts on your research and funding, as well as a few other topics.

The questionnaire is short and should take no more than ten minutes of your time.

**Fig 4. Representation appeal experiment in COVID-19 Survey Wave 2.**

in surveys by explicitly asking them to "express your personal views," whereas the community-representation condition explicitly encouraged them to "represent the scientific community." Figs 4–6 shows the operationalization in the Public Trust in Science Survey. Full invitation emails for each survey are available in S3–S5 Appendices in Supporting Information.

In the COVID-19 Survey Wave 4, a two-by-two experimental design was implemented, combining the two representation appeal conditions with two levels of information detail (none vs. some information). Fig 7 shows the experimental design, and full invitation emails are included in S6 Appendix (Fig 8).

```
Subject Line: Survey [Randomized (1) self-representation appeal:"regarding your views" / (2)
community-representation appeal: " ") on public trust in science

Dear Dr. [Surname only]:

I am writing to ask for your help with a national survey of scientists and engineers concerning
the [Randomized (1) self-representation appeal: "your personal view on factors that influence the
public's trust in science. You are part of a random sample that has been chosen to complete a
brief questionnaire. Your opinion and experiences are needed to inform further understanding
about this important topic." / (2) community-representation appeal: "scientific community's
views on factors that influence public trust in science. You are part of a random sample that has
been chosen to represent the scientific community by completing a brief questionnaire
concerning this important topic."]

The questionnaire is short and should take no more than ten minutes to complete.
```

**Fig 5. Representation appeal experiment in public trust in science survey.**

```
Subject Line: [Randomized (1) self-representation appeal: "Your" / (2): community-
representation appeal: "Scientific Community "] views on women's reproductive healthcare

Dear Dr. [Surname only]:

I am writing to ask for your help with a national survey of scientists and engineers concerning [
Randomized (1) self-representation appeal:"your personal views on women's reproductive
healthcare. You are part of a random sample that has been chosen to complete a brief
questionnaire concerning this important topic." / (2) community-representation appeal:"women's
reproductive healthcare. You are part of a random sample that has been chosen to represent the
scientific community by completing a brief questionnaire concerning this important topic."]

The questionnaire is short and should take no more than ten minutes to complete.
```

**Fig 6. Representation appeal experiment Women's Health Survey.**

**Randomization.** STEM academic scientists in the sample were randomly assigned to experimental conditions using random numbers generated in MS Excel. For the information appeal experiments, one-third of the sample was assigned to each of the three conditions (no information, some information, detailed information). For the representation appeal experiments, the sample was evenly split between self-representation and community-representation appeals. In the two-by-two design for the COVID-19 Survey Wave 4, the sample was divided equally among the four condition combinations.

**Measurement.** The outcome variable in each experiment is the survey response rate, defined as the proportion of responses within the eligible experimental sample. Responses included both completed and partial responses (coded as "1"), while nonresponses (coded as "0") included explicit refusals, surveys opened with no answers, or no reply to the invitation. The eligible experimental sample excluded ineligible STEM academic scientists (e.g., deceased, retired, or out of academia) and those unreachable during the survey period (e.g., on rotations or leave).

To assess randomization quality, we conducted balance tests for all six experiments using Pearson's Chi-squared test, accounting for demographic variables and prior SciOPS survey experience. Balance tests showed no significant differences in demographic variables (gender, academic field, academic rank) across treatment groups at a 0.05 significance level, confirming the experimental conditions were well-balanced. Additionally, some sampled scientists in the COVID-19 Survey Wave 4 had consented to become SciOPS panel members in 2022, indicating their greater

Subject Line: Long-Term Impact of COVID-19 on [Randomized (1) self-representation appeal "Your" / (2): community-representation appeal: "Scientific"] Research

Dear Dr. [Surname only],

How did COVID-19 [Randomized (1) self-representation appeal: **"impact your research**?" / (2) community-representation appeal: "**impact scientific research in the US"**] Please help answer this question by completing a very brief – less than 10 minutes – questionnaire to share [Randomized (1) self-representation appeal: "your personal pandemic experience" / (2) community-representation appeal: "pandemic experience"].

[Randomized (A) no information appeal: "" /

(B) some information appeal: "As you know, multiple studies have documented the numerous immediate effects of the pandemic on scientists like you. Less attention has been given to the longer-term impacts, the topic that we focus on here and are requesting your help with. In particular, we are interested in your general reflections on the ways that your COVID-19 experiences may have influenced your funding, research, and overall productivity over the past four years."]

You can access the online questionnaire by entering the following unique URL into your browser.

**Fig 7. Two-by-two experiment in COVID-19 Survey Wave 4.**

willingness to represent the scientific community and contribute to science communication, but no significant differences were found in the proportion of SciOPS panel members across experimental groups. Similarly, for the Vaccine Survey, while some respondents had participated in prior SciOPS surveys, no significant differences were found in this subgroup across experimental conditions. Detailed results are available in S2–S4 Tables in the Supporting Information.

## Ethical considerations

A consent form appeared on the first page of each survey instrument, informing participants of their rights to participate voluntarily or decline at any time. Participants were asked to indicate their consent by clicking to begin the survey. We informed, obtained, and documented consent through emails and the survey software systems. Survey participation was recorded as consent, while refusals, ineligibility, and non-contact were also documented through email communications. Given the nature of online surveys, the absence of foreseeable risks to participants, and the characteristics of our sample—busy academic scientists geographically distant from our institution—obtaining written consent was time-consuming and not feasible. Instead, we employed implied consent, a widely accepted and ethically appropriate approach for minimal-risk online survey research [90]. Research has found no substantial differences in how well online versus written consent informs participants [91], and requiring signed consent forms may reduce response rates and deter prospective participants who would otherwise be willing to complete the survey [92,93]. Study procedures were approved by the Arizona State University Institutional Review Board (ASU Study #00012476).

## Analytical strategies

Our statistical analysis follows a three-stage approach. First, for surveys with three experimental conditions, we conducted one-way analysis of variance with pairwise comparisons of response rates using the Tukey HSD test. For surveys with two experimental conditions, we used t-tests to compare response rates across conditions. Second, we utilized logistic regression to assess the effect of experimental conditions on the probability of survey participation controlling for covariates.

Third, we performed subgroup analyses to examine response rate differences by academic rank and prior interactions with SciOPS, including previous survey invitations and panel membership.

These subgroup analyses were based on several factors. First, STEM scientists at different academic ranks may face varying competition, promotion pressures, and time availability, potentially influencing their response to information and representation appeals. Second, prior SciOPS interactions could shape respondents' sensitivity to these appeals. Those invited to previous surveys may have pre-existing perceptions of SciOPS, affecting their responses differently from those of first-time participants. Additionally, SciOPS panel members, being more prosocial and committed to science communication, may have different sensitivity to survey appeals.

## Results

### Information appeal experiment results and discussion

Hypothesis 1 posits that lengthy survey invitation emails with more information discourage STEM academic scientists from participating in surveys compared to invitations with less information. Table 2 presents the response rates for each experimental condition across surveys involving information appeals, revealing mixed results. In the Survey of Scientists' Perceptions of Surveys, invitation emails without any information significantly increased response rates compared to emails with greatest amounts of information about the survey topic (19.8 percentage points higher, $p < 0.01$). In contrast, the COVID-19 Survey Wave 4 shows the opposite trend: STEM academic scientists receiving emails with no information responded at lower rates than those receiving emails with some information, albeit at a borderline significance level (2.7 percentage points lower, $p < 0.1$).

The S5 Table in Supporting Information provides logistic regression results that align with these findings. In the Survey of Scientists' Perceptions of Surveys (Model 1 in S5 Table), both treatments—some information and much information—significantly reduced the probability of academic scientists responding. Compared to emails without any information about the purpose of the survey, emails with some information were 9.6 percentage points less likely to motivate responses ($p < 0.1$), while emails with much information were also less likely to motivate responses (17.4 percentage points, $p < 0.01$). Conversely, in the COVID-19 Survey Wave 4 (Model 3), STEM academic scientists receiving emails with some information were 2.8 percentage points more likely to respond than those receiving emails with no information ($p < 0.05$). The salience of information details is inconsistent across different surveys, offering mixed support for Hypothesis 1. A brief and concise invitation email with minimal survey details does not consistently guarantee more responses. One possible explanation is that the salience of information depends on several factors, including the polarization of the survey topic, respondents' career stage, and their prior experience with SciOPS. To explore these contingencies, we conducted subgroup analyses.

Table 3 further illustrates these subgroup discrepancies by rank. In the Survey of Scientists' Perceptions of Surveys, assistant professors and full professors responded at significantly higher rates to emails without information compared to those with much information (24.4 and 22.7 percentage points higher, respectively, both at borderline $p < 0.1$). In the Vaccine Survey, invitation emails with much information obtained significantly higher response rates from non-tenure-track researchers (26.9 percentage points higher, $p < 0.05$), compared to emails providing some information. Additionally, associate professors responded more to emails with some information about the survey than to emails with no information (21.0 percentage points higher, $p < 0.1$). In the COVID-19 Survey Wave 4, emails providing some information resulted in significantly higher response rates from associate professors (5.4 percentage points higher, $p < 0.05$), compared to emails with no information.

The elaboration likelihood model posits that individual attitudinal change results from two distinct routes of persuasion (central and peripheral), depending on their motivation and ability to process a message [94]. The central route occurs when individuals are highly involved with the issues and possess the time and cognitive resources to thoughtfully consider the merits of the information presented in support of an advocacy. The peripheral route, by contrast, occurs when

**Table 2. Response rate by information appeals.**

| Survey | No information condition | Some information condition | Much information condition | No information vs Some information difference | Some information vs Much information | No information vs Much information |
|---|---|---|---|---|---|---|
| Survey of Scientists' Perceptions of Surveys | 0.443 | 0.326 | 0.244 | 0.117 | 0.081 | **0.198***** |
| Vaccine Survey | 0.377 | 0.343 | 0.421 | 0.034 | −0.078 | −0.044 |
| COVID-19 Survey Wave 4 | 0.079 | 0.106 | – | **−0.027*** | – | – |

Note.

*p < 0.1,

**p < 0.05,

***p < 0.01.

**Table 3. Response rate by information appeals across subgroups of academic rank.**

| Survey | Rank | No information condition | Some information condition | Much information condition | No information vs Some information difference | Some information vs Much information | No information vs Much information |
|---|---|---|---|---|---|---|---|
| Survey of Scientists' Perceptions of Surveys | Non-tenure track researcher | 0.381 | 0.235 | 0.300 | 0.146 | −0.065 | 0.081 |
| | Assistant professor | 0.400 | 0.233 | 0.156 | 0.167 | 0.077 | **0.244*** |
| | Associate professor | 0.367 | 0.313 | 0.273 | 0.054 | 0.040 | 0.094 |
| | Full professor | 0.510 | 0.429 | 0.283 | 0.082 | 0.146 | **0.227*** |
| Vaccine Survey | Non-tenure track researcher | 0.386 | 0.241 | 0.510 | 0.146 | **−0.269**** | −0.124 |
| | Assistant professor | 0.364 | 0.268 | 0.417 | 0.096 | −0.149 | −0.053 |
| | Associate professor | 0.281 | 0.491 | 0.356 | **−0.210*** | 0.135 | −0.075 |
| | Full professor | 0.426 | 0.364 | 0.417 | 0.062 | −0.053 | 0.009 |
| COVID-19 Survey Wave 4 | Non-tenure track researcher | 0.069 | 0.102 | – | −0.033 | – | – |
| | Assistant professor | 0.101 | 0.104 | – | −0.003 | – | – |
| | Associate professor | 0.048 | 0.102 | – | **−0.054**** | – | – |
| | Full professor | 0.086 | 0.111 | – | −0.025 | – | – |

Note.

*p < 0.1,

**p < 0.05,

***p < 0.01.

individuals have low involvement or limited processing capacity. Attitude shift is a result of a simple cue, such as brief arguments or an attractive information source rather than scrutinizing message content [95].

In terms of personal relevance and involvement, the Survey of Scientists' Perceptions of Surveys elicits a lower level of relevance from STEM academic scientific community. It addresses issues that rarely impact STEM academic scientists' daily work and attracts limited attention from both the STEM academic scientific community and the general public. In

contrast, the Vaccine Survey and the COVID-19 Survey Wave 4 both address high-stakes, critical and polarized issues that greatly affect both societal outcomes and the well-being of the STEM academic scientific profession, thus increasing level of relevance.

Career stage shapes STEM academic scientists' information process capacity. Assistant professors and full professors are often overwhelmed with time-sensitive research responsibilities and heavier survey burdens compared to associate professors and non-tenure track researchers [96–98]. Assistant professors, under the pressure of the tenure clock in the early stages of their academic careers, face heavier workloads but have fewer resources for flexibility [96]. Full professors, on the other hand, juggle multiple tasks that demand significant time, including greater mentoring responsibilities, managing multiple research projects and grants, supervising laboratories, and engaging in professional services outside their institutions [97]. Conversely, non-tenure-track researchers and associate professors, with relatively more flexible time management and less survey burden, are more likely to have capacity to engage with detailed invitation emails.

Taken together, these differences suggest that assistant and full professors are more likely to rely on the peripheral route to process survey requests. When the survey topic is less polarized and personally relevant—the Survey of Scientists' Perceptions of Surveys—they are more responsive to concise, efficient invitations that reduce cognitive processing load. In contrast, non-tenure-track researchers and associate professors are more likely to perform central route processing. For more polarized and relevant topics such as the Vaccine Survey and the COVID-19 Survey Wave 4, they are more cautious about expressing opinions on contentious issues and utilize their cognitive resources to scrutinize detailed survey information to understand the study's background and purpose before deciding to respond.

Table 4 shows that response rate differences across experimental conditions are also contingent on respondents' prior communication experience with SciOPS. In the Vaccine Survey, the sample are all SciOPS panel members. Invitation emails with much information induce significantly higher response rates from STEM academic scientists who had not previously participated in SciOPS surveys (28.1 percentage points higher, p < 0.01) compared to emails with some information. According to the central route of the elaboration likelihood model, individuals who are receiving SciOPS survey invitations for the first time may lack familiarity with SciOPS's communication style and, thus, need to mobilize more information-processing capacity to comprehend the messages and access its credibility. As a result, the effect of more detailed information is significant. Those who have previously responded to SciOPS surveys are likely to have a familiarity effect. Their prior exposure to similar communications enables them to rely on simple cues, such as recognizable sender

**Table 4. Response rate by information appeals across subgroups of prior communication experience with SciOPS.**

| | Prior communication experience with SciOPS | No information condition | Some information condition | Much information condition | No information vs Some information difference | Some information vs Much information | No information vs Much information |
|---|---|---|---|---|---|---|---|
| Vaccine Survey | Have been invited to SciOPS survey | 0.351 | 0.349 | 0.373 | −0.003 | 0.024 | 0.021 |
| | Have not been invited to SciOPS survey | 0.481 | 0.322 | 0.603 | 0.159 | **−0.281***** | −0.122 |
| COVID-19 Survey Wave 4 | SciOPS panel member | 0.315 | 0.472 | – | **−0.157*** | – | – |
| | Non SciOPS panel member | 0.064 | 0.083 | – | −0.019 | – | – |

Note.

*p < 0.1,

**p < 0.05,

***p < 0.01

and branding, to respond to the survey through a peripheral route. This result is consistent with previous findings about online survey panel members, which suggests that returning survey participants often may overlook aspects of the survey request from a known source and not scrutinize the message in depth [99,100].

For those sampled for the COVID-19 Survey Wave 4, all were invited to be SciOPS panel members in 2022 before this survey, but only some of them consented to join. This group of scientists demonstrates a strong commitment to supporting the scientific community and science communication through survey participation. Subgroup analysis shows that emails with some information resulted in significantly higher response rates from those SciOPS panel members (15.7 percentage points higher, p < 0.1), compared to emails with no information. According to the central route of the elaboration likelihood model, SciOPS panel members have a stronger relevance with SciOPS surveys compared to non SciOPS panel members. Their stronger commitment and relevance prompts them to more carefully consider the context and background of the surveys they participate in. Surveys with more information are more likely to attract their participation as this is more likely to provide them with sufficient information to evaluate the salience of the survey.

### Representation appeal experiment results and discussion

Hypothesis 2 posits that invitation appeals emphasizing self-representation motivate STEM academic scientists to participate in surveys more effectively than emails encouraging them to represent the scientific community. Table 5 compares response rates across experimental conditions for surveys with representation appeals, also revealing mixed results. In the Women's Health Survey, a self-representation appeal significantly increased response rates compared to a community-representation appeal, with a 9.5 percentage point higher response rate (p < 0.05). The S6 Table in Supporting Information presents logistic regression results, which show consistent findings while controlling for covariates. In the Women's Health Survey (Model 6), STEM academic scientists receiving community-representation appeals were 9.7 percentage pointsless likely to respond to the survey compared to those receiving self-representation appeals (p < 0.05). The results from the Women's Health Survey thus support Hypothesis 2.

Table 6 shows that in the Women's Health Survey, non-tenure-track researchers responded at significantly higher rates to a self-representation appeal compared to a community-representation appeal (31.1 percentage points, p < 0.01). Table 7 shows that the effect of self-representation appeals on response rates are not contingent on respondents' prior communication experience with SciOPS.

The results of the representation appeal experiments suggest that STEM academic scientists' inclination to represent their personal views vs. those of the scientific community depends on the context in which they are asked to express their opinions. It is only significant in the Women's Health Survey. This survey was conducted several months after the Supreme Court overturned *Roe v. Wade*, a time when women's reproductive health was a polarizing and highly debated issue. In such a contentious environment, STEM academic scientists were more likely to express their personal views

**Table 5. Response rate by representation appeals.**

| Survey | Self-representation Condition | Community-representation Condition | Response rate difference |
|---|---|---|---|
| COVID-19 Survey Wave 2 | 0.164 | 0.143 | 0.021 |
| Public Trust on Science Survey | 0.226 | 0.229 | −0.003 |
| Women's Health Survey | 0.400 | 0.305 | **0.095**** |
| COVID-19 Survey Wave 4 | 0.094 | 0.091 | 0.003 |

*Note.*

*p < 0.1,

**p < 0.05,

***p < 0.01.

**Table 6. Response rate by representation appeals across subgroups of academic rank.**

| Survey | Rank | Self-representation condition | Community-representation condition | Difference |
|---|---|---|---|---|
| COVI-19 Survey Wave 2 | Non-tenure track researcher | 0.103 | 0.081 | 0.022 |
| | Assistant professor | 0.196 | 0.188 | 0.008 |
| | Associate professor | 0.162 | 0.125 | 0.037 |
| | Full professor | 0.164 | 0.142 | 0.022 |
| Public Trust on Science survey | Non-tenure track researcher | 0.251 | 0.193 | 0.058 |
| | Assistant professor | 0.200 | 0.247 | −0.047 |
| | Associate professor | 0.216 | 0.240 | −0.024 |
| | Full professor | 0.235 | 0.222 | 0.013 |
| Women's Health Survey | Non-tenure track researcher | 0.587 | 0.276 | **0.311**\*\*\* |
| | Assistant professor | 0.297 | 0.229 | 0.068 |
| | Associate professor | 0.333 | 0.310 | 0.023 |
| | Full professor | 0.379 | 0.344 | 0.035 |
| COVID-19 Survey Wave 4 | Non-tenure track researcher | 0.097 | 0.069 | 0.028 |
| | Assistant professor | 0.121 | 0.087 | 0.034 |
| | Associate professor | 0.074 | 0.079 | −0.005 |
| | Full professor | 0.091 | 0.104 | −0.013 |

*Note.*

*$p < 0.1$,

**$p < 0.05$,

***$p < 0.01$.

**Table 7. Response Rate by representation appeals across subgroups of prior communication experience with SciOPS.**

| | Prior communication experience with SciOPS | Self-representation condition | Community-representation condition | Difference |
|---|---|---|---|---|
| COVID-19 Survey Wave 4 | SciOPS panel member | 0.453 | 0.333 | 0.119 |
| | Non SciOPS panel member | 0.071 | 0.075 | −0.004 |

*Note.*

*$p < 0.1$,

**$p < 0.05$,

***$p < 0.01$.

rather than represent perspectives of the scientific community. Results from subgroup analysis further show that non-tenure-track researchers were more likely to represent themselves than academic scientists with tenure or in tenure-track.

This pattern is aligned with self-categorization theory [101], which suggests that individuals with weaker group identification are more likely to affirm their individual identity and disidentify from the group when the group is under threat [102]. Non-tenure-track researchers are low group identifiers due to their institutional marginalization and identity insecurity within the academic scientific community, where holding a tenure-track position is a strong marker of group membership. In a polarized environment, the academic scientific community is internally highly divided and externally threatened, particularly under public scrutiny over its predisposition on contentious social issues. For individuals with insecure or marginalized identities, affiliating with the academic scientific community can feel risky. Thus, non-tenure-track researchers are more comfortable with expressing their personal opinions rather than representing a potentially controversial group.

As time passes since the initial COVID-19 outbreak, the declining salience of this survey topic may have influenced the effectiveness of self-representation and community-representation appeals differently in the COVID-19 Wave 2 and Wave 4 surveys. However, a comparison of experimental results from the two waves of the COVID-19 survey (Tables 4–6) show no significant differences in response rates between these appeals in either survey. This suggests that the effectiveness of self-representation and community-representation appeals was not dependent on changes in the salience of the COVID-19 survey topic. It is possible that by the time the COVID-19 Survey Wave 2 was conducted in the summer of 2021—more than a year after the outbreak—the topic salience may have already diminished for STEM academic scientists. By 2024, when the COVID-19 Survey Wave 4 was administered, any further change in topic salience was likely minimal.

## Discussion

Findings from six experiments reveal no linear or simplistic conclusion regarding whether the amount of information provided or type of appeal made might be differentially effective in increasing academic STEM scientists' survey response rates. Instead, the evidence highlights that both less and more information can lead to higher response rates under some conditions. Meanwhile, egoistic self-representation does not always lead to significantly higher response rates, compared to more altruistic community-representation appeals. These experimental conditions interact in complex and dynamic ways with survey response behavior. Table 8 summarizes the key findings across different surveys and subgroup analyses.

The heterogeneous effects of information and representation appeals on STEM academic scientists' participation depends on survey topic, career stage, and prior interactions with survey institutions. For survey topics, appeals with less information tend to increase participation for less polarized or socially discussed topics, while appeals with more information are preferred for highly critical or politicized issues. Self-representation appeals are more effective for polarized and trending topics. Regarding career stages, STEM academic scientists with limited time availability (e.g., assistant professors) or high information burdens (e.g., full professors) are more likely to prefer short, concise invitation email messages. Weak group affiliation for non-tenure-track researchers makes them prioritize self-representation when responding to contentious survey topics. Lastly, the relationship between recipients of survey invitations and senders influences the effectiveness of these appeals. For one, a familiarity effect often emerges in panel survey settings where recipients have received repeated survey requests from a known source [99,100]. Over time, as recipients become accustomed to the style and content of these invitations, they may rely less on the specific details within each request when deciding whether to participate. In such cases, the necessity for carefully crafted information and representation appeals may be diminished. For another, respondents with commitment to carefully expressing their opinions will require clear guidance and more details about the survey to evaluate their capacity and eligibility for participation.

**Table 8. Results summary for information appeal experiment.**

| Academic ranks/ Survey topics | High relevance and polarization | Low relevance and polarization |
|---|---|---|
| Non-tenure track researcher | Much information > Some information; Self-representation > Community-representation | N.S. |
| Assistant professor | N.S. | No information > Much information |
| Associate professor | Some information > No information | N.S. |
| Full professor | N.S. | No information > Much information |
| First time survey recipients | Much information > Some information | N.S. |
| Survey panel member | Some information > No information | N.S. |

*Note. N.S. denotes no significant results.*

Survey researchers are encouraged to tailor invitation emails on a case-by-case basis, as there is no one-size-fits-all formula for designing effective appeals. Strictly adhering to a specific level of detail or emphasizing a particular type of representation is unlikely to yield optimal results. Table 8 suggests that survey administrators should consider the configuration of survey topics and recipient characteristics when using information and representation appeals. First, for polarized topics, we recommend using self-representation appeals, particularly when recipients have low affiliation with the group and when the group faces external threats. Second, for topics with lower relevance to recipients, we suggest providing minimal information, particularly for survey recipients who lack time and capacity to engage deeply with the survey request. In contrast, for highly relevant and polarized topics, we recommend providing a higher level of detail to recipients with the capacity to process more information. Third, when administering surveys to a panel, we suggest sending a high-level informational invitation to recipients with strong commitment and a middle-level informational invitation to first-time recipients.

This study has several limitations. First, some nonrespondents may have made their decisions based solely on the email subject line. Due to ethical considerations and technical constraints, we could not track whether survey recipients opened the email and were exposed to the experimental treatments embedded in the email body. To address this, we manipulated the representation appeal condition in both the subject line and email content, ensuring exposure to experimental treatment regardless of whether the email was opened. Thus, the estimated effect of the representation appeal is not biased by the email open rate. In contrast, the information appeal condition could not be meaningfully conveyed in a brief subject line. It is possible that the unmanipulated subject line may have influenced survey recipients' propensity to open email messages compared with a hypothetical subject line manipulating the information appeal. Nevertheless, it does not bias the estimated effect of the information appeal on response propensity, as all experimental groups received the same subject line and the rates at which emails are opened can be assumed to be randomly distributed across experimental groups.

Second, while we tested our hypotheses across multiple random experiments, the sample sizes varied significantly, ranging from several hundred to several thousand participants. The lack of statistically significant results in some experiments may be attributed to smaller sample sizes and limited statistical power. Third, STEM academic scientists' personal interests in different survey topics may have influenced their attention to the content of the invitation email [99,103]. Although we use randomized trials and controlled for academic specialty in logistic regression models, we conducted an additional sensitivity check by examining field-specific response rates. Specifically, we analyzed responses to the Women's Health Survey and the Vaccine Survey under the assumption that faculty in public health might have greater interest in these two survey topics. We found that public health faculty who received invitations with greater information were significantly more likely to respond—by approximately 15 percentage points ($p < 0.1$), compared to those who received no information. However, we did not observe significant differences across other subgroups (see S7 Table in Supporting Information). These findings suggest that academic field may only partially capture topic interest. We recommend that future research develop more direct measures of personal interest in survey topics and explore how such relevance influences how recipients process and respond to survey invitations.

Despite these limitations, this study contributes to three lines of research. First, it is the first in the science communication literature to examine how invitation emails influence the participation of a unique population—STEM academic scientists—in survey research studies. As science communication becomes increasingly prominent in both society and academia, this work helps advance our understanding of available communication strategies to encourage STEM academic scientists to share their opinions and experiences in survey research. Unlike surveys targeting the general population, conducting surveys with STEM academic scientists is more challenging due to their time constraints and relatively small population size. Our study shows that designing tailored information and representation appeals can, under the right circumstances, provide a cost-effective approach to increasing STEM academic scientists' willingness to participate in research.

Second, this study provides new evidence on how the content of survey invitation messages impacts response rates. While existing research has explored the effects of egoistic appeals, altruistic appeals, topic salience, and other messaging strategies [48,70,74,79,99], we investigate two novel appeals: the volume of information provided and the framing of representation as self or community. Our findings highlight more possibilities to customize survey invitations to suit targeted populations. Additionally, results show that the effectiveness of these appeals is influenced by factors such as survey topics, recipients' prior interactions with the survey administrators, and their time availability and cognitive processing capacity. Our findings offer insights for previous research that reports mixed evidence on the effectiveness of specific strategies in influencing response rates. These variations in effectiveness can be attributed to additional factors related to survey design and the characteristics of the targeted populations.

Third, this study expands the survey response rate literature by examining a special, highly educated population: STEM academic scientists. Previous research has focused on highly educated groups such as teachers, health care providers, and politicians [44,63,75]. By introducing the scientific community into this body of work, we contribute to a broader understanding of response behavior among specialized professional populations.

This research opens several agendas for future study. First, we recommend additional research to replicate our experiments and examine how the effects of information and representation appeals vary across broader academic communities and other populations. Our findings and conclusions are limited to STEM academic scientists at U.S. R1 universities and may not generalize to academic scientists in other fields, scientists working in industry and government, international scholars, or the public in other professions. Scientists in non-STEM disciplines or industry settings have fundamentally different work environments, incentives, and communication norms, which may influence their reaction to information and representation appeals in ways that differ from STEM academic scientists. For example, social scientists, who more frequently engage in survey research, may interpret survey appeals differently than STEM faculty. Moreover, the salience of detailed information may vary across industry and government scientists, other professions (e.g., teachers, lawyers, or journalists), or the general public due to distinct information-processing preferences, occupational routines, and time availability. Similarly, the representation appeal experiment could be extended to other populations with strong social identity and connections (e.g., neighborhoods, ethnic groups, immigrants, and LGBTQ communities).

Second, future research could explore how information and representation appeals impact response quality, including breakoff rates, speeding, and straightlining. Providing more detailed information may help respondents consider questions more thoroughly, connect their answers to the survey's context, and offer higher-quality answers. Meanwhile, refined community-representation appeals could exert greater moral or altruistic pressure, motivating respondents to engage more carefully with surveys.

Third, it is worth examining whether information and representation appeals in survey invitations influence how individuals respond to survey questions. Variations in survey requests, question wording, and item ordering are known to shape respondents' interpretation of questions and their choices of answers [103–105]. In particular, representation appeals—asking individuals to represent themselves versus their professional association or community—may lead to differing responses to the same questions. Understanding how these appeals frame responses can help survey researchers better tailor invitation wording to align with their research goals, whether focused on capturing individual opinions or community perspectives.

## Supporting information

**S1 Table. Number of randomly selected institutions for sampling scientists.**
(PDF)

**S1 Appendix. Invitation emails template for survey of scientists' perceptions of surveys (information appeal experiment).**
(PDF)

**S2 Appendix.  Invitation emails template for vaccine survey (information appeal experiment).**
(PDF)

**S3 Appendix.  Invitation emails template for COVID-19 Wave 2 Survey (representation appeal experiment).**
(PDF)

**S4 Appendix.  Invitation emails template for public trust survey (representation appeal experiment).**
(PDF)

**S5 Appendix.  Invitation emails template for Women's Health Survey (representation appeal experiment).**
(PDF)

**S6 Appendix.  Invitation emails template for COVID-19 Wave 4 Survey (two by two – information and representation appeals experiment).**
(PDF)

**S2 Table.  Balance test results for information appeal experiments.**
(PDF)

**S3 Table.  Balance test results for representation appeal experiments.**
(PDF)

**S4 Table.  Balance tests results for COVID-19 Survey Wave 4.**
(PDF)

**S5 Table.  Logit models results of information appeal experiment.**
(PDF)

**S6 Table.  Logit models results of representation appeal experiment.**
(PDF)

**S7 Table.  Field-specific sensitivity check for vaccine survey and Women's Health Survey.**
(PDF)

## Author contributions

**Conceptualization:** Tipeng Chen, Timothy P. Johnson, Eric W. Welch.

**Data curation:** Tipeng Chen.

**Formal analysis:** Tipeng Chen.

**Funding acquisition:** Eric W. Welch.

**Investigation:** Tipeng Chen, Jinghuan Ma, Ashlee Frandell, Lesley Michalegko.

**Methodology:** Timothy P. Johnson.

**Project administration:** Jinghuan Ma, Ashlee Frandell, Lesley Michalegko.

**Writing – original draft:** Tipeng Chen, Timothy P. Johnson.

**Writing – review & editing:** Tipeng Chen, Timothy P. Johnson, Jinghuan Ma, Ashlee Frandell, Lesley Michalegko, Eric W. Welch.

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
