## [Decision Letter · Decision Letter 0]

Dear Dr. Chen,

Thank you for submitting your manuscript to PLOS ONE. After careful consideration, we feel that it has merit but does not fully meet PLOS ONE’s publication criteria as it currently stands. Therefore, we invite you to submit a revised version of the manuscript that addresses the points raised during the review process.

We look forward to receiving your revised manuscript.

Kind regards,

Marco Improta

Academic Editor

PLOS ONE

Journal Requirements:

3. In the online submission form, you indicated that to safeguard privacy and maintain confidentiality of participants, the data is available from author upon request.

Reviewers' comments:

Reviewer's Responses to Questions

**Comments to the Author**

1. Is the manuscript technically sound, and do the data support the conclusions?

Reviewer #1: Partly

2. Has the statistical analysis been performed appropriately and rigorously?

Reviewer #1: Yes

3. Have the authors made all data underlying the findings in their manuscript fully available?

Reviewer #1: No

4. Is the manuscript presented in an intelligible fashion and written in standard English?

Reviewer #1: No

Reviewer #1: Revise the discussion to better integrate mixed findings with theory, emphasizing contextual moderators (e.g., topic polarization, career stage).

Clarify practical recommendations with specific examples or decision trees for survey designers.

Address generalizability by acknowledging limitations and suggesting future research directions (e.g., cross-disciplinary or international replication).

Improve readability by streamlining the methods section and adding a visual summary of key findings (e.g., a conceptual diagram).

**Do you want your identity to be public for this peer review?** For information about this choice, including consent withdrawal, please see our Privacy Policy

Reviewer #1: **Yes: ** Ekramul Islam

---

## [Author Response · Author response to Decision Letter 1]

27 May 2025

The cover letter and response to reviewers are provided as PDF attachments.

---

## [Editor Report · Decision Letter 1]

Invitation Appeals and STEM Academic Scientists Research Participation: Findings from Six Survey Experiments

PONE-D-25-11539R1

Dear Dr. Chen,

We’re pleased to inform you that your manuscript has been judged scientifically suitable for publication and will be formally accepted for publication once it meets all outstanding technical requirements.

Kind regards,

Marco Improta

Academic Editor

PLOS ONE
---

## [Editor Report · Acceptance letter]

PONE-D-25-11539R1

PLOS ONE

Dear Dr. Chen,

I'm pleased to inform you that your manuscript has been deemed suitable for publication in PLOS ONE. Congratulations! Your manuscript is now being handed over to our production team.

Kind regards,

on behalf of

Dr. Marco Improta

Academic Editor

PLOS ONE